# A Study of the Literature on Intrauterine Treatment Options for Chronic Placental Insufficiency with Intrauterine Growth Restriction Using Intrauterine Intravascular Amino Acid Supplementation

**DOI:** 10.3390/life13061232

**Published:** 2023-05-23

**Authors:** Lisa van Uden, Michael Tchirikov

**Affiliations:** University Clinic of Obstetrics and Prenatal Medicine, Center of Fetal Surgery, University Medical Center Halle (Saale), Martin Luther University Halle-Wittenberg, Ernst-Grube Strasse 40, 06120 Halle (Saale), Germany

**Keywords:** IUGR, intrauterine growth restriction, treatment, intrauterine treatment, port, fetal amino acids, intraumbilical infusion, cordocentesis

## Abstract

Background: Intrauterine growth retardation (IUGR) is a very serious prenatal condition with 3–5% incidence of all pregnancies. It results from numerous factors, including chronic placental insufficiency. IUGR is associated with an increased risk of mortality and morbidity and is considered a major cause of fetal mortality. Currently, treatment options are significantly limited and often result in preterm delivery. Postpartum, IUGR infants also have higher risks of disease and neurological abnormalities. Methods: The PubMed database was searched using the keywords “IUGR”, “fetal growth restriction”, “treatment”, “management” and “placental insufficiency” for the period between 1975 and 2023. These terms were also combined together. Results: There were 4160 papers, reviews and articles dealing with the topic of IUGR. In total, only 15 papers directly dealt with a prepartum therapy of IUGR; 10 of these were based on an animal model. Overall, the main focus was on maternal intravenous therapy with amino acids or intraamniotic infusion. Treatment methods have been tested since the 1970s to supplement the fetuses with nutrients lacking due to chronic placental insufficiency in various ways. In some studies, pregnant women were implanted with a subcutaneous intravascular perinatal port system, thus infusing the fetuses with a continuous amino acid solution. Prolongation of pregnancy was achieved, as well as improvement in fetal growth. However, insufficient benefit was observed in infusion with commercial amino acid solution in fetuses below 28 weeks’ gestation. The authors attribute this primarily to the enormous variation in amino acid concentrations of the commercially available solutions compared with those observed in the plasma of preterm infants. These different concentrations are particularly important because differences in the fetal brain caused by metabolic changes have been demonstrated in the rabbit model. Several brain metabolites and amino acids were significantly decreased in IUGR brain tissue samples, resulting in abnormal neurodevelopment with decreased brain volume. Discussion: There are currently only a few studies and case reports with correspondingly low case numbers. Most of the studies refer to prenatal treatment by supplementation of amino acids and nutrients to prolong pregnancy and support fetal growth. However, there is no infusion solution that matches the amino acid concentrations found in fetal plasma. The commercially available solutions have mismatched amino acid concentrations and have not shown sufficient benefit in fetuses below 28 weeks’ gestation. More treatment avenues need to be explored and existing ones improved to better treat multifactorial IUGR fetuses.

## 1. Introduction

According to the current state of development and research in the field of obstetrics and prenatal medicine, (extreme) prematurity is the main cause of neonatal mortality and morbidity. Both birth weight and gestational age are the most important determinants [1,2]. In recent years, due to advances in research and development in neonatal intensive care, the survival of preterm infants has improved significantly and the limit of viability has been lowered from the 22nd to the 24th week of gestation. However, this survival is associated with a high rate of complications [3]. Few of these extreme preterm infants do not suffer any subsequent complications [3]. While 40% of all preterm infants below the 28th week of gestation die within the first five years [4], 91% of all preterm infants at the 23rd week of gestation and 67% of preterm infants at the 24th week of gestation die within that same time period, that is, the first five years [5]. Rarely do extremely preterm infants survive without severe late effects [6]. Among premature babies born in the 22nd week of gestation, 95 to 96% show pronounced physical and/or mental damage [5]. These preterm infants have severe retinopathy almost 90% of the time, and 42% are affected at 23 weeks’ gestation. Only less than 20% of extremely preterm infants below the 24th week of gestation survive without developing necrotizing enterocolitis, sepsis, meningitis, bronchopulmonary hypoplasia, pronounced cerebral hemorrhage and/or other severe complications [5].

An additional risk factor for preterm birth as well as for deficiency births is intrauterine growth retardation (IUGR), which significantly increases neonatal mortality. Intrauterine growth retardation (IUGR) results from many different factors, including chronic placental insufficiency, and is a very serious prenatal condition. The incidence of IUGR is approximately 3–5% of all pregnancies [7]. Although intrauterine growth restriction is associated with an increased risk of mortality and morbidity and can be considered a major cause of fetal mortality (approximately 25% to 40% of all intrauterine fetuses that die are IUGR fetuses) [8], once diagnosed, prenatal treatment options are limited [9].

As a major organ in transport and metabolism, the placenta has a vital role in the nutrition and metabolism of the fetus and provides oxygen and nutrients. In this context, the placenta is a kind of sensor for nutritional, metabolic, endocrine and vascular conditions and fetal needs [10]. In placental insufficiency, active placental transport of amino acids, glucose, and oxygen from the mother to the fetus is inhibited [11,12,13]. Impaired trophoblast invasion during the embryonic period results in altered formation of fetal vascularization in the feto-maternal unit and inadequate transformation of maternal spiral arteries with concomitant reduced uterine perfusion. At the present stage of research, a number of changes in the activity of amino acid transporters have been identified in an IUGR placenta [14]. In 2021, Rosario’s team [15] also found that reduced placental amino acid transport precedes fetal growth restriction in a non-human primate experiment. In this experimental group with maternal nutrient restriction, a 40% reduction in amino acid transport activity was noticed, which in the course resulted in an IUGR pregnancy. Significantly lower fetal and maternal amino acid concentration differences have been found in IUGR pregnancies compared to term pregnancies [12,14]. Maternal concentrations of most amino acids were significantly higher than in normal pregnancies, with significantly lower amino acid concentrations in IUGR fetuses. A reduction in placental transport of the essential amino acids, leucine [16] and threonine [17], in vivo has already been demonstrated in the sheep placental insufficiency model. A reduction in the ratio of leucine and phenylalanine between fetal and maternal accumulation in vivo was also demonstrated in human IUGR pregnancies [18].

In these IUGR fetuses, there is usually a redistribution of venous and arterial blood flow to maintain adequate supply to the fetal brain. The redistribution reduces the blood supply to the fetal liver [19], which also leads to a reduced production of important proteins, lipids, carbohydrates and growth factors [20,21,22,23] and consequently to a reduced cell proliferation of the fetal organs [22].

(Doppler) sonographic examination reveals this frequent redistribution of arterial and venous blood flow [19,24,25] and marked intrauterine growth retardation. Here, the flow characteristics of the blood in the vessels and the growth retardation of the fetus are primarily assessed. For example, in severe IUGR, reduction in cerebral artery blood flow resistance and pulsatility index (PI) in the middle cerebral artery predicts an 11-fold increased risk of intraventricular hemorrhage, periventricular leukomalacia, hypoxic ischemic encephalopathy, necrotizing enterocolitis, bronchopulmonary dysplasia, sepsis, and death [26]. The time between diagnosis of brain sparing and delivery in severe IUGR fetuses has been reported to range from two to fifteen days [26].

Postpartum, infants with IUGR also have an increased risk of long-term neurologic deficits and for poorer cognitive performance [27], metabolic syndrome [28,29], premature puberty, and short stature compared with gestational-aged children, suggesting an impact on fetal programming [28,29]. The fetal brain in particular has been shown to be vulnerable to IUGR conditions [30,31], as the fetal brain does not show the steepest growth until between 29 and 36 weeks of gestation [32]. Notably, extremely preterm infants show severe neurodevelopmental disorders in about 10–15% during infancy and have mild motor, behavioral, and learning disorders in about 30–40% of cases at school age [9]. An essential amino acid for neuronal development is tryptophan [33], which can cause hyperexcitability and anxiety due to reduced transplacental transfer. Because of this dysfunctional neurodevelopment, a high rate of aggressive behavior has been seen in IUGR baboons [34]. 

Cooke [35] reported that cognitive development at eight years of age was related to intrauterine growth. In this regard, birth weight below the tenth percentile was associated with poor growth, developmental deficits, and language problems at 56 months of age. Pregnancy prolongation has been shown to improve these neurological abilities in IUGR fetuses, as demonstrated in term infants at 17 years of age by Pat et al. [36].

Presently, direct treatment of trophoblast invasion in the embryonic period is not possible. With the current diagnostic tools, it is not yet possible to distinguish between physiological and abnormal at this stage of the embryonic period. In the later course of pregnancy, it is not possible to completely replace such a complex organ as the human placenta. Part of the treatment of an IUGR fetus may be fetal amino acid and glucose supplementation, which could lead to prolonged gestation and increased fetal growth [9,36,37,38,39,40]. Application of amino acids and glucose to the amniotic fluid was tried as early as the 1970s, but increased rates of amniotic infection syndromes and preterm labor thwarted this treatment method [32,41]. The goal was to increase amino acid concentrations in fetal plasma to treat IUGR fetuses.

In an additional way, the aim is to avoid the permanent serious complications of premature birth associated with fetal programming. Patients with an IUGR pregnancy are delivered early usually by an incisional delivery to avoid intrauterine amniotic death due to the deficiency supply resulting from placental insufficiency. In spite of all this, the only established treatment for placental insufficiency at this time is preterm delivery, which in itself increases perinatal mortality and morbidity [7,42,43,44] and therefore has to be discussed.

## 2. Materials and Methods

The literature search was conducted in PubMed for the period between 1975 and 2023. The PubMed database was searched using the keywords “IUGR”, “fetal growth restriction”, “treatment”, “management” and “placental insufficiency”. These terms were also combined together.

## 3. Results

In total, there were 4160 papers, reviews and articles dealing with the topic of IUGR. Among those, therapeutic options of IUGR were described in 1022 papers, wherein a majority discussed the diagnosis and the pre- and postpartum management, especially in terms of monitoring. In terms of direct therapeutic options, cesarean section was predominantly mentioned in these.

Only 495 papers from 1975 onward dealt with intrauterine therapy in the broadest sense, with the majority focusing on therapy of a possible underlying disease (medication, maternal underlying disease, drug use, genetics, etc.). 

There were only a few papers (15 papers) which directly dealt with a prepartum therapy of IUGR, and 10 of these were based on an animal model. Overall, the main focus was on maternal intravenous therapy with amino acids or intraamniotic infusion.

In 1971, J. Dudenhausen et al. [41] applied an amino acid solution once in six cases to mature unborn children during birth after opening of the bladder via a vaginally inserted intrauterine catheter and determined the concentrations of leucine, valine, and tyrosine in the plasma of the mother and umbilical cord blood at several time intervals. The aim was to detect the uptake of amino acids applied intrauterine. The research group was able to detect a significant increase in concentration in fetal plasma as early as 0.5 h after amino acid infusion. The maximum amino acid content could be detected after 1.5 to 2 h. It was proven that amino acids given intraamnially are accepted by fetuses. 

As early as 1986, E. Saling et al. [32] implanted a catheter into the fetal abdomen through which a transperitoneal solution of premature infant formula, 0.9% NaCl solution, and trace elements were infused. This model by E. Saling is called peritoneal dialysis, in which the peritoneum is also permeable to corpuscular components. In the case described, weight gain and a marked increase in the amount of amniotic fluid were observed within nine days of infusion therapy. Post the ninth day, birth occurred after a rupture of the membranes. To minimize the risk of infection, the catheter was tunneled over approximately 15 cm under the skin of the pregnant woman and connected to a subcutaneously implanted delivery capsule. Fetal bowel injury and dislocation of the catheter from the fetal abdomen were not described. 

In 2002, a maternal intravenous infusion of amino acids was performed by S. Ronzoni et al. [45] in eight of eighteen IUGR pregnancies to increase fetal amino acid concentrations. The infusion was given in the maternal fasting state before elective caesarean delivery, at which time maternal, umbilical cord venous and arterial blood samples were later collected. In the IUGR group with the maternal infusions, all amino acid concentrations were found to be significantly elevated in the mother. Increased uptake of some amino acids by the fetus was demonstrated after the maternal amino acid infusion. In the umbilical cord vein, only the concentrations of valine, methionine, isoleucine, leucine, phenylalanine, arginine, serine, glycine and proline were increased. However, a change in the uptake of the three essential amino acids lysine, histidine and threonine and the non-essential amino acid alanine could not be shown [46]. As early as 1998, Jansson et al. [47] showed reduced placental transport of lysine in IUGR placentas. Lysine is an essential amino acid and an important nutrient for the fetus.

M. Tchirikov et al. [9,37,48] described direct delivery of amino acids and glucose to IUGR fetuses via a subcutaneously implanted intravascular perinatal port system as illustrated in Figure 1 and shown in Appendix A. One goal was to compensate for the deficiency supply as much as possible and to improve fetal growth with significant prolongation of pregnancy.

In 2010, M. Tchirikov et al. [9] described the implantation of a subcutaneously implanted intravascular perinatal port system in a human fetus with IUGR and uterine artery resistance. Daily infusions of amino acid solution and 10% glucose were given for approximately four weeks. There was significant improvement in fetal weight and a constant increased resistance to flow in the uterine arteries. Pregnancy was prolonged to 37 + 0 completed weeks of gestation. After delivery, the catheter showed a correct intravascular position as shown in Figure 2. At six-month follow-up, there were no abnormalities compared with children without a history of IUGR. Tchirikov et al. [9] suggest normalization of fetal programming and concomitant prevention of chronic diseases associated with IUGR. 

In a 2017 prospective pilot study, Tchirikov et al. [37] described implantation of subcutaneously implanted intravascular perinatal port systems in six IUGR fetuses with “brain sparing” with supplementation of amino acids and glucose and compared with a control group of eight IUGR fetuses. One patient also received the combination with daily hyperbaric oxygenation. The study showed a significant prolongation of pregnancy duration and higher weight gain in the experimental group. The study described an inadequate benefit of infusion with commercial amino acid solution in fetuses under 28 weeks of gestation. With supplementation of amino acid concentration, increased amino acid imbalance was observed in the IUGR fetuses, which led the authors to not recommend the use of the commercial amino acid solution for the extremely preterm IUGR fetuses because, according to the authors, the commercially available AA solution has enormous deviations from the AA levels observed in the plasma of preterm infants.

Hyperbaric oxygenation applied was to prevent oxygen toxicity to the placenta and fetal brain. 

In 2018, Tchirikov et al. [48] published a case report in which one patient received a subcutaneously implanted intraumbilical perinatal port system with daily amino acid and glucose infusion in combinations with hyperbaric oxygenation (HBO, 100% O_2_) and the second patient received hyperbaric oxygenation alone. Here, HBO was intended to prevent possible development of lactic acidosis and to increase oxygen diffusion in the placenta. Both patients had preeclampsia in addition to severe IUGR. In the first case, the patient received nine days of infusion with an amino acid solution and seven days of parallel hyperbaric oxygenation. At 5 years of age, the child showed delayed speech development without other neurological disorders. In the second case at 25 + 0 completed week of gestation, intraumbilical administration of amino acids was omitted because, according to the authors, an amino acid solution with appropriate fetal amino acid concentration was not available and the commercial AA solution showed no benefit before 28 weeks of gestation [37] HBO was performed for one day. Delivery of the extremely preterm infant occurred a short time later due to late fetal decelerations. The preterm infant died a few days later after pulmonary hemorrhage.

Already in 2013, E. van Vliet et al. [49] described the changes in the fetal brain due to metabolic changes in a rabbit model. Most brain metabolites (N-acetylaspartylglutamic acid (NAAG), N-acetylaspartate (NAA), and pyroglutamic acid) and amino acids (ornithine, L-lysine, aspargine, histidine, and the leucine intermediate 2-keto-isovalerate) were significantly decreased in IUGR brain tissue samples. Because amino acids are significant for brain growth, metabolism, and function, abnormal neurodevelopment occurred with decreased brain volume among other effects.

## 4. Discussion

The intrauterine growth retardation is a multifactorial severe disease for which there is currently no sufficient treatment. Due to its high relevance with an incidence of about 3–5% of all pregnancies [7] and about 40% of all stillbirths [8], it is altogether essential to establish further studies and possibilities of treatment. Current studies are mainly related to the compensation of the nutrient deficiencies existing due to chronic placental insufficiency resulting in growth retardation and the normalization of fetal programming to prevent chronic diseases.

Intrauterine growth restriction is often associated with placental growth restriction, which can be caused by a deficiency in nutrition associated with hypoxia and ischemia. Placental growth may be restricted in favor of fetal growth for the time being [50]. It remains to be seen how far this placental growth restriction has a role as a diagnostic tool and what effect the supplementation of an intravascular amino acid solution also has on placental growth and maturation. Moreover, it is questionable how much a possible placental growth restriction can limit fetal growth and development even under supplementation of an amino acid solution. However, all studies show a very low number of cases in animal and human experiments. In 2013, E. van Vliet et al. [49] described 16 rabbits including 6 rabbits in the control group in the rabbit model. Additionally, the studies by different teams around M. Tchirikov described one patient in 2010, a total of 14 patients including 8 control cases in 2017, and 2 patients in 2018. All these limited number of cases show the difficulties and also the carefulness with regard to intrauterine therapies, since any intervention can be dangerous and can result in the death of the already abnormal fetus, and are therefore difficult for ethical and practical purposes. This is evidenced, among other things, by the termination of the 1970s study due to the increased rate of amniotic infection syndromes [41]. S. Ronzoni et al. [45] also pointed out in 2002 that it is not possible to infer all IUGR pregnancies because of the small number of cases in the various studies, since IUGR pregnancies have a heterogeneous nature.

Currently, it is impossible to replace such a complex organ as the human placenta in order to counteract intrauterine growth retardation in chronic placental insufficiency. The alterations related to placental tissue angiogenesis, impaired syncytial and cytotrophoblast development, decreased proliferation, and altered trophoblast invasion [11,12,13] cannot currently be treated, thus forming an irreversible disorder. Therefore, current treatment options and studies focus on the consequences and mechanisms of chronic placental insufficiency [13,51,52]. An important goal here is the prolongation of pregnancy. Paz et al. demonstrated as early as 2001 that term infants with a history of fetal growth restriction were not at an increased risk for low intelligence scores at 17 years of age [36]. 

Another goal is to compensate for the deficiency supply of the fetus. Currently, amino acid supplementation is part of several studies in which different amino acid infusions are delivered in different ways to the pregnant woman or directly to the fetus. Because active placental transport of amino acids and glucose to the fetus is reduced in IUGR fetuses [19,43], supplementation of these is a logical treatment option. Amino acid supplementation, among other things, suppresses fetal protein degradation rates, promotes build-up rates, and thus effectively supports fetal growth [53]. The addition of amino acids could also potentially improve the other signaling pathways and impairments. Because the fetal brain in particular is susceptible to deficiency under IUGR conditions [30,31] and has been associated with changes in neurotransmitter profiles [54], a reduction in total brain volume, and regional changes in gray and white matter volume [55], amelioration of this deficiency should be associated with improved neurological development. Of course, the side effects of amino acid supplementation also need to be further investigated. It has been suggested that such addition stimulated the production of lactate and alanine in fetal muscle [56] and thus may also induce lactic acidosis. 

It is absolutely necessary to develop a better understanding of the different nutrient transports in order to improve our knowledge of the mechanisms of altered fetal growth. In addition, it is essential to research further the placental metabolism of glucose, lipids and amino acids in order to develop potentially preventive strategies.

There is also a need to develop a good and tolerable amino acid solution for intrauterine continuous infusion. As Tchirikov described in 2017 [37], the most commonly used infusions currently have a different composition of amino acids compared to the physiologically occurring amino acid concentration in timely developed fetuses. The commercial AA solutions used in the protocol did not contain aspartic acid, glutamic acid, glutamine, ornithine (Fresenius) and cysteine, ornithine, glutamine, and taurine (B. Braun), which show a huge deviation, according to the authors, from the physiological AA proportions observed in the plasma of extreme preterm infants (Figure 3) [42,57,58]. In addition, it is worth noting that especially the amino acids asparagine and taurine are significantly increased in the ratio between mother and fetus, that is, by more than three times. Additionally, the amino acids lysine, ornithine and aspartate are significantly increased by more than two times [46]. Amino acids are nutrients, regulators of gene expression, protein phosphorylation cascade, cell signaling molecules, and hormone synthesis [38].

Thus, long-term supplementation of commercial amino acid solutions in IUGR fetuses may result in an imbalanced plasma amino acid ratio, which may lead to poorer short- and long-term outcomes. Due to these differences, in 2018, Tchirikov [48] refrained from supplementation in a severe case of IUGR because an amino acid solution with appropriate fetal amino acid concentration was not available according to the authors and the commercial AA solution showed no benefit before 28 weeks of gestation [37]. The working group of S. Ronzoni et al. [45] also assumes a worse outcome in extremely preterm infants when amino acid supplementation is given that does not correspond to natural fetal concentrations. Current amino acid concentrates failed to increase fetal concentrations or diplacental uptake of some essential amino acids. Therefore, it can be assumed that the concentrates do not provide all amino acids necessary to increase fetal protein synthesis.

It has been shown in some studies that continuous amino acid and glucose infusion through the umbilical vein by a subcutaneously implanted port system has improved fetal growth and prolonged pregnancy by some weeks [9]. Nevertheless, the authors have already discussed the necessity of researching and preparing an amino acid solution that matches the natural fetal concentration [37,45]. The accurate amino acid combination in the solutions is essential to improve the effect, and therefore the success of the treatment. At this time, no available appropriate amino acid solutions that are similar to the fetal physiological amino acid concentration or concentration ratios in fetal blood plasma in utero are also available to benefit fetuses before the 28th week of gestation. 

There is also an economic benefit in providing adequate treatment for these IUGR fetuses, which could reduce the burden on the health care system. The neonatal therapy of these fetuses is often very costly due to the long and very complex care in the intensive care unit. Additionally, (sequelae) diseases, such as metabolic syndrome, type II diabetes mellitus, increased risk of stroke, and coronary artery disease, have a chronic course [9] and may require lifelong treatment. Fetuses with a history of IUGR have a higher mortality rate, more perinatal complications [59], and sequelae such as impaired visual, auditory, and executive functions, developmental delays, psychiatric and behavioral problems, cardiovascular and pulmonary disorders, and resulting socioeconomic burdens [4,59,60,61,62]. All of these conditions and sequelae could potentially be reduced or prevented by treating IUGR during the intrauterine period. The economic benefit of this treatment is, of course, not the ultimate medical goal and should in no way be the rationale for research. However, this could make it interesting to promote studies from the economic side in the long term, which in turn may influence the willingness and promotion of these studies.

Many other factors, such as the influence of sex-specific characteristics of the placentas and their responses to the adverse intrauterine environment, have not yet been adequately explored. IUGR pregnancy in itself has a very multifactorial etiology. It is also worth questioning how maternal habits in terms of physical activity, qualitative and quantitative caloric intake, and ethnicity influence IUGR conditions. In an experiment with non-human primates, Rosario et al. [15] already made the hypothesis that maternal nutrient restriction precedes reduced placental amino acid transport with the development of an IUGR fetus [15], which has significant relevance, especially in developing countries. All of these factors could significantly determine the development and emergence of IUGR fetuses and thus need to be further assessed in future research.

Since before the year 2000, amino acid concentrations from fetal cord blood have been analyzed and documented in many studies. As already shown, the recently approved infusion solutions for parenteral nutrition for preterm infants include an amino acid concentration that does not match the fetal intrauterine concentration ratio (Figure 3). As preterm infants and especially IUGR fetuses show insufficient intrauterine growth and development, they need a special supply to ensure adequate growth and nutrition. Therefore, research of a specific amino acid solution is rather important to allow intrauterine therapy as well as to allow sufficient parenteral nutrition of preterm infants.

In conclusion, further and new treatment options need to be explored in the future. IUGR pregnancies is multifactorial and includes alterations in fetal blood oxygen levels, acid–base balance, production of anabolic and catabolic hormones, and activation of stress pathways, which should also be investigated in future prenatal intervention strategies. Each study and new therapeutic approach will improve the understanding of the causes, effects, and possible treatments of chronic placental insufficiency with intrauterine growth retardation to better treat this condition in the future.

Subcutaneously implanted intravascular perinatal port system in combination with continuous amino acid infusion could be a sufficient treatment option in human fetuses with IUGR. Long-term nutrient infusion into the umbilical vein could allow prolongation of gestation and improved growth in IUGR fetuses. At the same time, restoration of fetal physiological concentrations of amino acids, glucose, trace elements, hormones, growth factors, and vitamins in IUGR fetuses could improve abnormal fetal programming. Because of the relatively few IUGR cases studied and the heterogeneous nature of IUGR pregnancy, generalizing the present results to all IUGR pregnancies is highly problematic at present. Therefore, in the future, not only appropriate amino acid infusions need to be developed, but also more cases of intrauterine treatment of IUGR fetuses need to be investigated consecutively. An amino acid infusion that is similar to the fetal physiological amino acid concentration or concentration ratios in fetal blood plasma in utero may also benefit preterm infants. The preterm infants are not similar developmentally to those born on term and could better meet their individual nutrient needs with the help of this infusion adapted to them and may also improve their development ex utero.

## Figures and Tables

**Figure 1 life-13-01232-f001:**
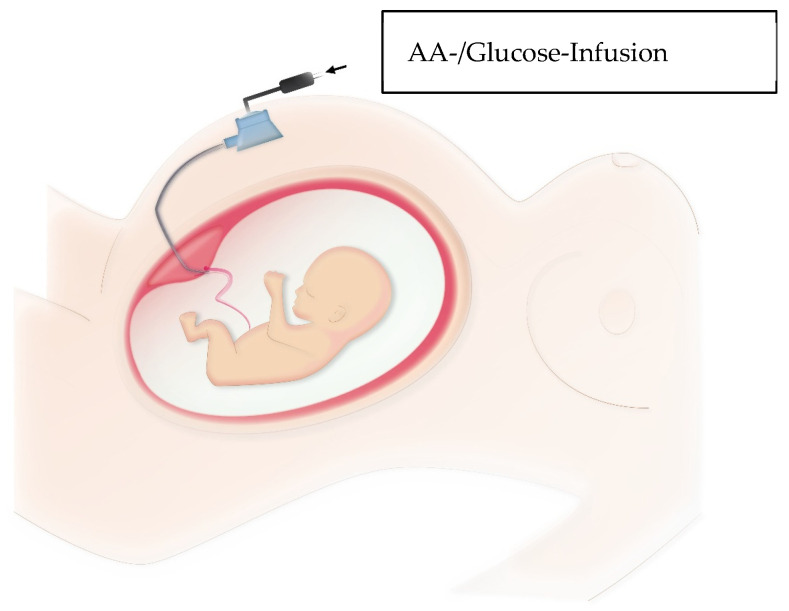
Method of subcutaneously implanted intravascular perinatal port system. PakuMed medical products GmbH, Germany, CE 0481; Tchirikov Patent US-9839767-B2) Picture from own collection.

**Figure 2 life-13-01232-f002:**
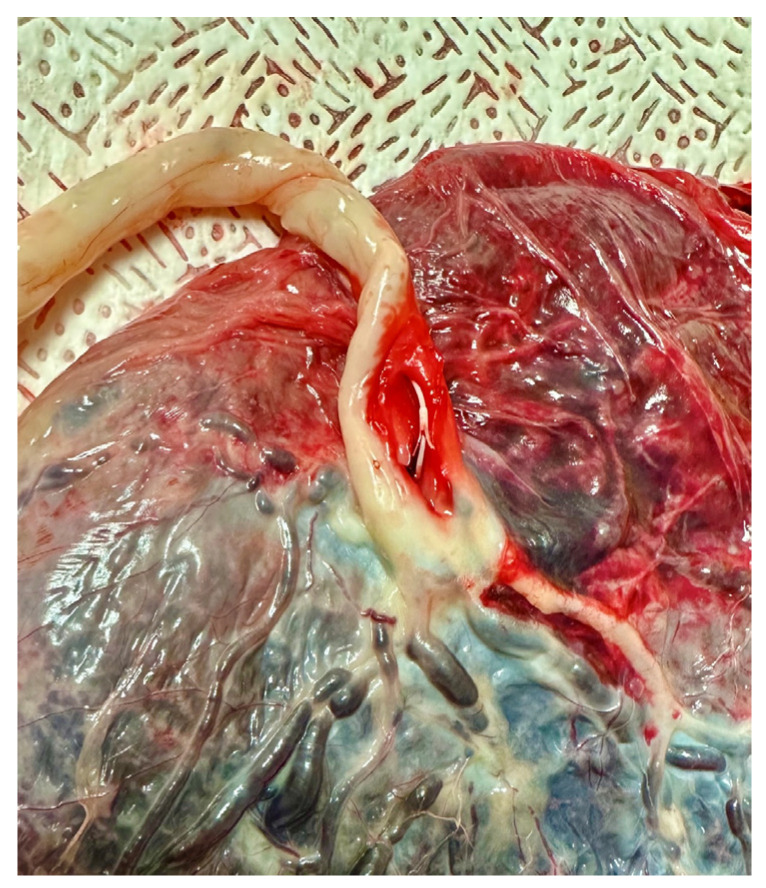
Placenta with catheter inserted into the umbilical vein with a small anchor system (to prevent dislocation, PakuMed medical products GmbH, Germany, CE 0481; Tchirikov Patent US-9839767-B2) Picture from own collection.

**Figure 3 life-13-01232-f003:**
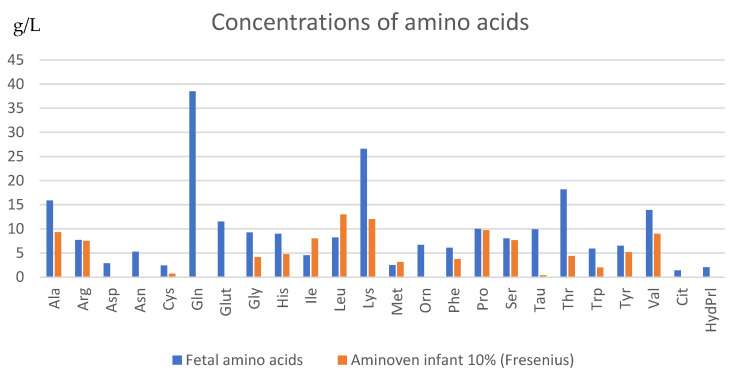
Concentrations of amino acids in the conventional amino acid infusion compared to the adapted specific amino acid infusion (×500) [13].

## Data Availability

No new data were created or analyzed in this study. Data is contained within the article or Appendix A.

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
