# Peer review of "A Study of the Literature on Intrauterine Treatment Options for Chronic Placental Insufficiency with Intrauterine Growth Restriction Using Intrauterine Intravascular Amino Acid Supplementation"

_life, 2023, doi:10.3390/life13061232_

Round 1
Reviewer 1 Report
The authors submitted a relevant review dealing with the complex scientific hot topic of IUGR. The review is well written and relevant for both, the specific scientific community as well as for innovative clinical and teaching approaches, respectively.
I suggest to accept the paper.
Author Response
Thank you for your review.
Reviewer 2 Report
This manuscript 'Intrauterine treatment options for chronic placental insufficiency with intrauterine growth restriction using intrauterine intravascular amino acid supplementation' is an important overview over published literature to find fetal treatment of FGR and to decrease morbidity and IUFD by better placenta function. It is a reminder for further research.
some comments and questions:
1. lines 76-78: '....... due to altered angiogenesis of placental tissue and insufficient trophoblast invasion' - could you please re-formulate this? Please differentiate fetal vascularization and maternal vascularization in the feto-maternal unit.
2. line 124: 'Direct cause treatment of chronic placental insufficiency is not possible at this time, as an organ as complex as the human placenta cannot be replaced.....' - no doubt about that, but suggested treatments influence maternal and fetal vascularization and so placental function....so, probably replace this sentence.
3. I would like to recommend to add to the title: 'Intrauterine treatment options for chronic placental insufficiency with intrauterine growth restriction using intrauterine intravascular amino acid supplementation, a study of literature.
4. Consider that Fetal FGR is often associated with placental growth restriction in your discussion ....
Author Response
Thank you for your review.
